# The Route of Stress in Parents of Young Children with and without Autism: A Path-Analysis Study

**DOI:** 10.3390/ijerph182010887

**Published:** 2021-10-16

**Authors:** Elisa Leonardi, Antonio Cerasa, Rocco Servidio, Angela Costabile, Francesca Isabella Famà, Cristina Carrozza, Letteria Spadaro, Renato Scifo, Sabrina Baieli, Stefania Aiello, Flavia Marino, Gennaro Tartarisco, David Vagni, Giovanni Pioggia, Liliana Ruta

**Affiliations:** 1Institute for Biomedical Research and Innovation (IRIB), National Research Council of Italy, 98164 Messina, Italy; elisa.leonardi@istitutomarino.it (E.L.); antonio.cerasa@irib.cnr.it (A.C.); francescaisabella.fama@istitutomarino.it (F.I.F.); cristina.carrozza@istitutomarino.it (C.C.); liaspad@yahoo.it (L.S.); stefania.aiello@irib.cnr.it (S.A.); flavia.marino@irib.cnr.it (F.M.); gennaro.tartarisco@irib.cnr.it (G.T.); david.vagni@irib.cnr.it (D.V.); giovanni.pioggia@irib.cnr.it (G.P.); 2S. Anna Institute and Research in Advanced Neurorehabilitation (RAN), 88900 Crotone, Italy; 3Pharmacotechnology Documentation and Transfer Unit, Preclinical and Translational Pharmacology, Department of Pharmacy, Health Science and Nutrition, University of Calabria, 87036 Arcavacata, Italy; 4Department of Cultures, Education and Society, University of Calabria, 87036 Arcavacata, Italy; rocco.servidio@unical.it (R.S.); angela.costabile@unical.it (A.C.); 5Centre for Autism Spectrum Disorders, Child Psychiatry Unit, Provincial Health Service of Catania (ASP CT), 95124 Catania, Italy; renato.scifo@aspct.it (R.S.); sabrina.baieli@aspct.it (S.B.)

**Keywords:** autism, stress, parents, anxiety, depression, structural equation modelling

## Abstract

We provide a conceptual model on the complex interaction between stress, psychological predisposition, and personality traits, accounting for gender, in parents of children with and without autism. We performed a path analysis using a structural equation modeling approach in a sample of parents including 60 ASD and 53 TD couples. In parents of typically developing children (TD), depression level and age are the main direct predictors of stress through the mediating effect of anxiety. Otherwise, in the ASD parent group, the personality trait ‘openness’ directly predicts the defensive response and stress levels without the mediating effect of anxiety. Our data suggest a route of action in promoting new behavioral strategies to prevent parenting stress, making families run smoothly.

## 1. Introduction

Autism spectrum disorder (ASD) is a highly heterogeneous condition with a broad variety of developmental trajectories, symptoms severity, and functional impairment throughout life [1,2,3]. Therefore, individuals with ASD typically need lifelong support, and caregivers must face a daily care load associated with stressful family routine [4,5,6,7], continuous psychological adjustments [8,9], financial difficulties, and heightened emotions [10,11]. Parental stress is the experience of distress that arises from the needs associated with the parenting role [12]. This condition occurs when the family is unable to restore functioning following the introduction of a stressor by engaging in normal family coping strategies [13,14].

Several studies have observed increased psychological stress in families of children with autism as compared to parents of children with Down syndrome [15,16], attention-deficit/hyperactivity disorders [17], intellectual disability [18,19], or typical development [17]. In fact, evidence from the literature suggests that ASD is the condition that produces the highest parental stress as compared to other neurodevelopmental conditions [6,20,21]. Higher levels of stress, anxiety, and depression have been consistently reported in parents of children with ASD [22,23]. Psychological stress and burnout in parents of children with ASD have been associated with depression, anxiety [24], and decreased family cohesion [25,26], and it appears to reduce caring abilities [27]. All these factors contribute to a significantly lower parental well-being [28], satisfaction, and quality of life [17,28,29].

Furthermore, ASD parents have shown a higher incidence of psychopathology [30,31,32,33,34]. This evidence is partially related to the child’s autism condition, but it has also been linked to the specific neuropsychological profile of parents [35] in terms of personality traits and ability to cope with stressful events. 

Specifically, subclinical autistic features have been found in parents of children with autism and reported as “Broader Autism Phenotype” (BAP) [36,37,38]. The BAP has been associated with personality characteristics [39] such as rigidity, detachment, and hypersensitivity to criticism [36,40,41] as well as interpersonal difficulties [36]. In some studies, the expression of the BAP was also negatively correlated with Big Five Personality traits such as extraversion and conscientiousness and positively correlated with neuroticism [42]. Interestingly, the presence of a BAP in ASD parents has been correlated with stress and depression [43,44] and in turn with the risk of developing anxiety [44,45]. Conversely, states of depression and anxiety [46], maladaptive coping styles, and feelings of guilt have been related to levels of psychological stress within families of children with autism [47,48,49,50]. Sociodemographic characteristics such as parental age, educational levels, and lower income have also been associated with higher parental stress [43,51,52,53,54].

Although the reciprocal correlations between stress and specific measures of psychological well-being have been previously addressed, to the best of our knowledge, the complex interplay between factors that predict and mediate stress levels such as anxiety, mood, and personality traits, with sociodemographic characteristics as a control variable, in ASD parents has never been explored. This specific focus represents the main novel contribution of the present study that aims to investigate which kind of mediation occurs among all these factors. For this reason, we performed a path analysis using a structural equation modeling (SEM) approach in a large group of parents of children with autism compared with demographically matched parents of children with Typical Development (TD). In particular, we sought to explore if there are different combinations of factors as well as the role of specific mediating variables, predicting psychological stress in ASD parents with respect to TD.

## 2. Materials and Methods

### 2.1. Participants

A sample of 173 parent couples (82 ASD and 91 TD couples) of young children with and without autism aged between 3 and 6 years of age, both males and females (25% females in the ASD group and 51% females in the TD group) were enrolled in the study. Parents of children with autism were recruited and tested at the clinical facilities of the Institute for Research and Innovation in Biomedicine of the National Research Council of Italy (IRIB-CNR) in Messina and at the Centre for Autism Spectrum Disorders, Child Psychiatry Unit, Provincial Health Service (ASP-CT) in Catania, Italy. Parents of typically developing children (TD) were recruited in three different mainstream nursery schools and primary schools in Messina. Inclusion criteria in both the autism and TD parent groups were (1) being a native Italian speaker, (2) being aged between 25 and 55 years, and (3) being biological parents. Specific inclusion criterion in the autism parent group was having a child with a clinical diagnosis of ASD according to the Diagnostic and Statistical Manual of Mental Disorders, Fifth Edition (APA, 2013). In the TD parents, the group exclusion criteria were the following: (1) having a child with a clinical diagnosis of autism or other neurodevelopmental conditions (such as language and/or motor delay, ADHD, etc.); (2) family history of autism, intellectual disabilities, language delay in first degree relatives. In the ASD group, five couples of parents were excluded for the following reasons: three couples of parents were adoptive parents; two couples of parents initially agreed to participate but subsequently refused to contribute to the study. In the TD group, twelve parents were excluded for the following reasons: one parent (mother) was a foreigner, while one parent (father) presented a mild intellectual disability; two couples of parents had a first-degree relative with ASD; three couples of parents had a first-degree relative with intellectual disability. Furthermore, parents (*n* = 43 in total; *n* = 17 ASD and *n* = 26 TD parents) who did not complete more than 50% of items on each of the primary measures were excluded from the analyses. Therefore, a final sample of 60 ASD and 53 TD couples was analyzed in the study. In the ASD group, 56 couples were married, 3 were divorced, and 1 was a single mother. In the TD group, 51 couples were married, 1 couple was divorced, and 1 was a single mother.

The study received ethical clearance by the Ethics Committees of CNR (ethical clearance, 1 August 2018) and ASP-CT (Prot. N. 498), respectively, and all the caregivers signed an informed consent to participate in the study.

### 2.2. Procedure

Demographic information such as age, gender, education, and employment were gathered from all the participants and were used as covariates in our models. All parents completed the following neuropsychological assessment battery: (1)Parenting Stress Index, Short Form (PSI-SF). The PSI-SF is a 36-item, self-report measure designed to assess parental distress in parents of children aged between birth and 12 years [55]. Each item is measured using a Likert-type scale with values ranging from 1 (strongly disagree) to 5 (strongly agree). It contains a ‘Parental Stress Total Score’ which includes three subscales of 12 items each: Parental Distress (PD; e.g., “I feel trapped by my responsibilities as a parent”), Parent–Child Dysfunctional Interaction (PCDI; e.g., “Sometimes I feel my child doesn’t like me and doesn’t want to be close to me”), Difficult Child (DC; e.g., “My child makes more demands on me than most children”) and a Defensive Response domain, which includes 7 items (e.g., “I often have the feeling of not being able to cope very well with situations”). Defensive Response measures the level of social desirability of the parent, that is, the tendency to minimize a problematic parent–child relationship and to give a positive self-image.(2)State-Trait Anxiety Inventory-Form Y—Second Edition (STAI-Y2) [56] is a self-report questionnaire that contains 20 items for assessing trait anxiety and 20 items for state anxiety. Each item is measured using a Likert-type scale with values ranging from 1 (not at all) to 4 (very much so). In our study, we examined the Trait Anxiety domain, with the purpose of measuring the personological characteristics related to anxiety in parents of children with and without ASD. The total score in the Trait Anxiety domain of the STAI-Y2 ranges from 20 to 80, with scores >40 indicating above-average levels of trait anxiety.(3)Beck Depression Inventory (BDI-II) is a self-report questionnaire with 21 items assessing the presence and intensity of depressive traits based on the diagnostic criteria for depressive disorders. It gives a total score and two subscores related to the Somatic–Affective Area (loss of energy, altered sleep and appetite, agitation and crying, etc.) and to the Cognitive Area (pessimism, guilt, self-criticism). Cut-offs are the following: 0–13: minimal depression, 14–19: mild depression, 20–28: moderate depression, and 29–63: severe depression [57,58]. Answers are given using a four-point Likert-type scale ranging from 0 to 3.(4)Big Five Questionnaire (BFQ) to assess personality. The BFQ contains 5 domain scales: Extraversion/Energy (indicates the quality and intensity of interpersonal relationships, the level of activity of the subject as well as dynamism), Agreeableness (i.e., the way of relating to others, being kind, cooperative, friendly), Conscientiousness (refers to the capacity for self-regulation and self-control, being scrupulous, thoughtful, accurate), Emotional Stability (indicates the ability to manage stress, preserving one’s balance, without being overwhelmed by external events) and Mental Openness (i.e., proactive research, the pleasure of consulting what is unfamiliar, the interest in acquiring knowledge). For each of the Big Five domains, two subdimensions have been identified: Extraversion/Energy (E) includes Dynamism (Di) and Dominance (Do); Agreeableness (A) includes Cooperation/Empathy (Cp) and Friendliness/Friendship (Co); Conscientiousness (C) includes Scrupulousness (Sc) and Perseverance (Pe); Emotional Stability (S) includes Control of emotions (Ce) and Control of impulses (Ci); Mental Openness (M) includes Openness to culture (Ac) and Openness to experience (Ae). For each subdimension (consisting of 12 questions), half of the statements are formulated positively with respect to the scale name, while the other half are formulated negatively. In addition, there is a Lie (L) scale designed to measure a social desirability response set and the tendency to distort the meanings of the scores [59,60]. Therefore, in total, the BFQ is composed of 132 self-report items. The items are rated on a five-point Likert scale ranging from 1 (almost never true) to 5 (almost always true).(5)The Wechsler Abbreviated Scale of Intelligence (WASI-II) [61] is a standardized measure of cognitive intelligence and provides a Verbal Intelligence Quotient (gives an overall indication of the ability to understand, process, and organize the information presented in verbal form_VIQ), a Performance Intelligence Quotient (skills related to practical performances that involve understanding and organization of material to be processed in a perceptual and motor form_PIQ), and a Total Intelligence Quotient (general cognitive ability_TIQ). The WASI-II is useful for detecting cognitive abilities in clinical, educational, and research settings. The WASI-II is suitable for use with individuals between the ages of 6 and 90 and includes four subtests known as Block Design (BD), Vocabulary (V), Matrix Reasoning (MR), and Similarities (S).

Assessment was administered in a standardized way, keeping the same order of measures presentation in all the participants, and it was divided into two sessions. In the first session, the participants filled out the PSI-SF, STAI-Y2, BDI-II, and BFQ, while during the second visit, the WASI-II was administered by an expert clinical neuropsychologist (E.L.), specifically trained on the procedures of administration and scoring of the measure.

### 2.3. Data Analysis

Descriptive statistics were computed for each variable. The Kolmogorov–Smirnov test was carried out and confirmed the assumptions of normality for all variables. All variables were normally distributed, except for father/mother education. Bivariate Pearson’s correlations using a bootstrap sample of 5000 with 95% bias-corrected and accelerated (BCa) confidence intervals were computed among the variables of the study. Demographic and clinical differences between ASD and TD groups were tested using a two-sample t-test, a Mann–Whitney, and a chi-square test analysis. The internal reliability was obtained by computing the alpha of Cronbach (α). All analyses had two-tailed alpha levels of 0.01 for defining significance. The SPSS 25 package (IBM, NY, USA) was used to run the preliminary statistical procedures. 

A path analysis using structural equation modeling (SEM) was performed using Mplus 7.04 [62]. Path analysis is the simplest case of SEM, since we did not include latent variables. As recommended by Hu and Bentler [63], multiple indices were used to evaluate model fit (adopted cut-offs in brackets): the chi-square (χ^2^) test value with the associated *p* value (*p* > 0.05), comparative fit index (CFI ≥ 0.95), Tucker–Lewis Index (TLI ≥ 0.95), root-mean-squared error of approximation (RMSEA ≤ 0.06), and its 90% confidence interval, and standardized root mean square residual (SRMR < 0.08). Parameter estimation for the path-SEM model was performed with the maximum-likelihood parameter with standard errors and a mean-adjusted chi-square test statistic that were robust to non-normality (MLMV) as Maydeu-Olivares [64] suggested. The MLMV chi-square test statistic is also referred to as the Satorra–Bentler (S-B) chi-square.

The partial model and the full model were compared to determine the mediating role of anxiety in the relationship between the independent and the dependent variables. The models were assessed by using a chi-square (χ^2^) difference test, which indicates whether the constraints are justified [65]. Specifically, starting from the full model, which included all the control variables such as age, gender, education, and job, we removed step-by-step path coefficients not significant at the 5% level to obtain a more parsimonious model. Finally, only gender and age were included in the analyses as a control variable. We used GPower tool (http://www.gpower.hhu.de/; 02/21/2020 - Release 3.1.9.6) to make power analysis. With alpha = 0.01 and power = 0.8, the projected sample size needed for large effect size d = 0.8 is approximately *N* = 98 for group comparisons. Thus, our proposed sample size of 226 parents (120 ASD and 106 TD) should be adequate for the main objective of this study.

The main model aimed to test whether demographic factors, personality traits, and psychological measures impacted the perception and levels of stress in parents of children with and without ASD.

## 3. Results

### 3.1. Preliminary Analysis

Descriptive statistics are presented in Table 1 and Table 2. Generally, we found that autism and typical development parents are well matched in terms of age, education, and IQ (Table 1). Despite this similar picture, we found large significant differences among groups considering the psychological status (Depression and Anxiety), personality (Conscientiousness and Openness), and stress level (Perception of Stress and Defensive Response). Autism parents are characterized by higher anxiety, depression, and stress levels, as well as by lower conscientiousness and openness scores with respect to the typical development group. 

### 3.2. Path Analysis

The hypotheses of the study were investigated by using a path-SEM statistical approach where, initially, all data from the two groups were pooled together. The results of the partial mediating model, after controlling for age and gender, provided a good fit to the data, χ^2^S-B (4, *N* = 165) = 4.22, *p* = 0.238, CFI = 0.997, TLI = 0.973, RMSEA = 0.050, 90% CI (0.000, 0.149), SRMR = 0.017. However, only females, *β* = −0.273, showed higher levels of anxiety. In other words, gender predicted the levels of anxiety. In the next step, a full mediated model was tested by constraining all the direct relationships among the independent and the dependent variables. After comparing the two models (full and partial), we found that the larger model with more freely estimated parameters fits the data better than the smaller, Δχ^2^(12) = 30.77, *p* = 0.002; then, the partial mediating model was selected as the final one (Figure 1). 

Specifically, results demonstrated a negative direct effect of extraversion on both defensive response, *β* = −0.137, *SE* = 0.06, *p* = 0.025, and anxiety, *β* = −0.176, *SE* = 0.05, *p* = 0.001, as well as a negative correlation between emotional stability and anxiety, *β* = −0.275, *SE* = 0.06, *p* < 0.001. Conversely, we found a direct and positive relationship between depression and both defensive response, *β* = 0.263, *SE* = 0.08, *p* = 0.001, and stress *β* = 0.259, *SE* = 0.08, *p* = 0.002 and between depression, *β* = 0.584, *SE* = 0.06, *p* < 0.001 and anxiety, respectively. Finally, anxiety was related with both defensive response, *β* = 0.342, *SE* = 0.08, *p* < 0.001, and stress, *β* = 0.259, *SE* = 0.09, *p* = 0.003.

To investigate group differences, a second model was tested, by using the same constructs and relationships and including the profile of the parents (ASD and TD) as grouping variables. This model exhibited a good fit to the data, χ^2^S-B (6, *N* = 165) = 5.88, *p* = 0.437, CFI = 1.00, TLI = 1.00, RMSEA = 0.000, 90% CI (0.000, 0.141), SRMR = 0.015). As shown in Figure 2, results indicated similar effects across the two groups. However, in the ASD group, there was a direct effect of openness, *β* = −0.308, *SE* = 0.12, *p* = 0.008, and depression, *β* = 0.271, *SE* = 0.12, *p* = 0.020 on the defensive response. Additionally, openness was directly related to stress, *β* = −0.304, *SE* = 0.11, *p* = 007. Regarding the typical development group, depression was directly related to defensive response, *β* = 0.245, *SE* = 0.09, *p* = 0.008, and stress, *β* = 0.405, *SE* = 0.10, *p* < 0.001. Finally, gender held the same effects for autism and typical development groups, while age predicted only stress for the typical development group. 

Regarding the mediation analysis, we found that in the autism group, anxiety marginally mediated the relationship between depression and defensive response, *β* = 0.150, *SE* = 0.08, *p* = 0.060. Conversely, in the typical development group, we found that anxiety partially mediated the relationship between BDI and defensive response, *β* = 0.252, *SE* = 0.061, *z* = 4.155, *p* < 0.001, while anxiety fully mediated the relationship between emotional stability and defensive response, *β* = −0.218, *SE* = 0.056, *z* = −3.877, *p* < 0.001, and between extraversion and defensive response, *β* = −0.113, *SE* = 0.040, *z* = −2.803, *p* = 0.005.

Finally, the results of the mediational analysis in the whole sample (Table 3) indicated that the relationships between the independent and the dependent variables were partially mediated by anxiety. 

## 4. Discussion

In this study, we provide a new relationship model to shed new light on the complex interaction between stress, psychological, and personality traits, considering gender, in parents of children with and without autism. We confirm that raising a child with autism involves experiencing significantly higher levels of stress [66,67], anxiety, and depression [5,11,37,68] with respect to demographically matched parents of children with TD [46] (Table 2). Starting from this evidence, we used a path-SEM analysis to define the interplay between personality, anxiety, and mood status on stress levels, considering gender. Considering the pooled data, we found that some aspects of personality traits such as extraversion and emotional stability as well as depression and gender are the main relevant factors that predict the levels of stress in both groups of parents. It is noteworthy that the influence of these factors is mainly mediated by levels of anxiety. In particular, extraversion and emotional stability are negatively associated with the defensive response through the mediating effect of anxiety, while depression is positively associated with defensive response and stress both directly and through anxiety. These findings suggest that low scores in extraversion, emotional dysregulation, and depression are mainly related to a lack of control over psychological defensive responses and elevated distress. Anxiety exerts a pivotal role in this path model, consistently mediating the effect of predictive variables on stress domains. Furthermore, the effects of psychological and personality traits are also affected by gender, which exerts a direct influence on anxiety. Mothers display a significantly higher association with anxiety than fathers, and it in turn influences the effect on defensive response and stress. These findings are in line with recent evidence showing higher levels of stress in ASD mothers when compared to fathers in relation to parental emotion regulation, family functioning, and educational level [69].

When the model was applied separately to the two groups of parents, some specific relationships emerged. Indeed, in the autism parental group, the personality trait ‘openness’ directly predicted both the defensive response and stress levels, without the mediating effect of anxiety. Openness reflects, in part, a willingness to consider new ideas, as well as to question one’s values and beliefs. Therefore, considering the BAP hypothesis [38,70], we might expect a negative relationship between openness and parental stress. Our data would seem to confirm this hypothesis, suggesting that a lower level of openness in the group of ASD parents may be associated with a preference for familiar routines to new experiences and a narrower range of interests. Hence, it may be argued that the presence of personality traits linked to autism could be directly responsible for parental distress and a lack of defensive response to stress, increasing the risk for mental health problems. Conversely, in the typical development parent group, depression and age are the only main direct predictors of defensive response and stress levels.

Previous studies have reported the association of stress with mental health problems in parents of children with autism [9,46]. In line with our results, depression symptoms have been correlated with higher parenting distress especially in mothers underlying how withdrawing, loss of vital energy, feelings of victimism, sense of guilt, contextual amplification, and perceived disconnection contribute to experiencing elevated levels of distress [11,71,72]. Other studies have pinpointed the relationship between stress and anxiety in parents of children with autism [46], showing higher levels of anxiety and stress levels in mothers [73]. Personality traits such as neuroticism have been associated with higher levels of depression and lower levels of subjective well-being associated with stress, especially in mothers also in other clinical groups such as children with intellectual disabilities [74].

Shedding light on the interplay between sociodemographic features, personality, and psychological traits on parental stress may provide a unique opportunity for targeting family-focused interventions that are appropriate to the individual family burden and perceived parenting stress. The way parents respond to their children’s needs, in fact, is strictly dependent on their internal resources that may contribute to or protect against parenting stress [69].

### Limitation

The main limitation of our study is related to the characteristics of psychological assessment. Indeed, many questionnaires are self-reporting, thus relying heavily on participants’ ability to estimate their own personality, being truthful rather than giving the more socially acceptable answers. Furthermore, a substantial minority of questionnaires (12%) had to be excluded due to incomplete responses. Future research is needed to replicate our findings considering more objective measurement of psychological distress (i.e., cortisol measurements; neuroimaging markers).

## 5. Conclusions

Parental stress in families with a child diagnosed with autism is a frequent burdensome experience that deserves attention and intervention. Finding ways to moderate or mediate parenting stress will lead to better overall quality of life. Here, we describe the pathway that leads to parental distress starting from specific psychological and sociodemographic factors. Path-SEM analysis shows that there are different factors that influence the stress levels specifically in autism with respect to TD parents. In the former group, the personality trait openness and the defensive response are the only direct stress predictors, whereas in the TD parental group, depression, defensive response, and age are the main factors influencing stress perception. Our study highlights the importance of considering the complexity of interaction between personality traits, depressive symptoms, and anxiety to predict distress and burnout in parents of children with autism. Only by looking at the whole picture we will be able to identify individual factors that will help the parents of children with autism reduce their burden of distress and to improve the quality of their life.

## Figures and Tables

**Figure 1 ijerph-18-10887-f001:**
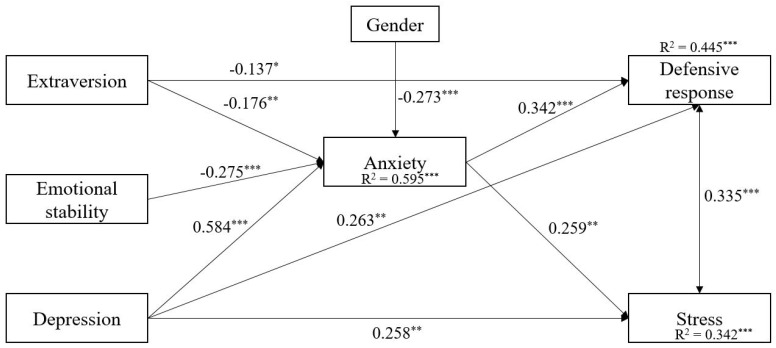
Results of the path model with standardized estimates in the whole sample. Note: * *p* < 0.05. ** *p* < 0.01. *** *p* < 0.001. Female = 1, Male = 2.

**Figure 2 ijerph-18-10887-f002:**
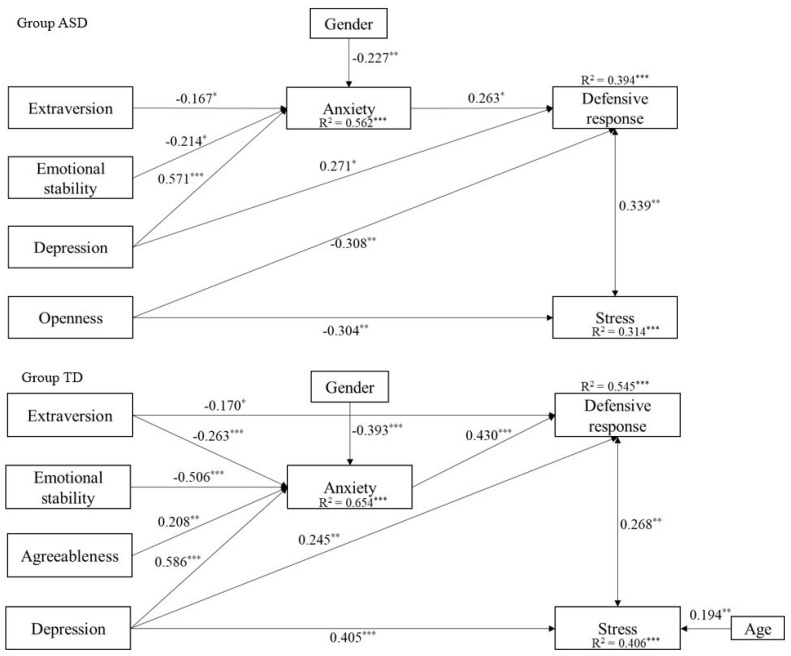
Results of the path model with standardized estimates in parents of children with autism (ASD) and typical development (TD) parents. Note: * *p* < 0.05. ** *p* < 0.01. *** *p* < 0.001. Female = 1, Male = 2.

**Table 1 ijerph-18-10887-t001:** Descriptive and neuropsychological characteristics in parents of ASD and TD children.

Variables	ASD	TD	*p*-Level
Age Mother	38.5 ± 4.5	39.8 ± 5.2	0.1 *
Age Father	42.1 ± 5.5	43.1 ± 5.1	0.2 *
Education Mother	14.87 ± 3.32	15.69± 3.39	0.21 *
Education Father	13.92 ± 4.24	14.88 ± 3.92	0.24 *
Job Mother Unemployed Administrative Professional Management Manual, technical Clerical Missing	26 (47.3%) 13 (23.6%) 9 (10.9%) 6 (10.9%) - 1 (1.8%)	14 (26.4%) 16 (30.2%) 7 (13.2%) 14 (26.4%) - 2 (3.8%)	0.06 ^ 0.58 ^ 0.62 ^ 0.07 ^ - -
Job Father Unemployed Administrative Professional Management Manual, technical Clerical Missing	3 (6.3%) 14 (29.2%) - 16 (33.3%) 14 (29.2%) 1 (2.1%)	- 4 (7.7%) 46 (88.5%) - - 2 (3.8%)	- 0.02 ^ - - -
Verbal IQ Mother	97.8 ± 8.9	98.8 ± 9.9	0.67 *
Performance IQ Mother	102.1 ± 12.5	101.7 ± 13.9	0.91 *
Total IQ Mother	100.1 ± 10.3	104.7 ± 22.7	0.3 *
Verbal IQ Father	97.9 ± 8.1	95.6 ± 8.3	0.31 *
Performance IQ Father	101.7 ± 12.2	105.7 ± 12.2	0.23 *
Total IQ Father	103.5 ± 23.8	104.4 ± 8.4	0.53 *

Data are given as mean values and standard deviation (SD) or median [range] in square brackets, as appropriate. * = Two samples *t*-test. ^ = Chi-square. ASD: autism spectrum disorders; TD: typical development; Education labels: 1 = primary school; 2 = secondary school; 3 = high school; 4 = bachelor’s degree; 5 = master degree; Job labels: Unemployed; Public/private worker; Technician/skilled workers; Public employees.

**Table 2 ijerph-18-10887-t002:** Psychological variables.

	ASD	TD	*t*-Value	*p*-Level	Cohen’s *d*
Mood Status					
*Depression (BDI-2)*	9.64 ± 7.8	5.98 ± 5.9	3.74	<0.001	0.53
*Anxiety (STAI-Y2)*	49.4 ± 10.2	44.6 ± 7.8	3.78	<0.001	0.52
Personality (BFQ)					
*Extraversion*	46.92 ± 8.8	48.66 ± 10.3	−1.28	0.201	0.19
*Agreeableness*	47.74 ± 10.4	50.92 ± 10.2	−2.18	0.031	0.31
*Conscientiousness*	49.76 ± 9.7	53.38 ± 8.2	−2.85	0.005	0.41
*Emotional Stability*	54.26 ± 9.7	55.50 ± 9.6	−0.91	0.364	0.12
*Openness*	40.04 ± 8.7	43.53 ± 9.8	−2.66	0.008	0.38
Stress					
*Defensive response*	57 ± 31.7	40.51 ± 31.1	3.53	0.001	0.53
*Stress*	49.24 ± 24.9	25.64 ± 21.8	6.62	<0.001	1.1

ASD = Autism spectrum disorders; TD = Typical development; Stress = Parental Stress Total Score. BDI: Beck Depression Inventory; STAI: State-Trait Anxiety Inventory-Form; BFQ: Big Five Questionnaire.

**Table 3 ijerph-18-10887-t003:** Testing the pathways of the mediational analysis with standardized estimates.

Pathway	Estimate	SE	*z*	*p*
Depression --> Defensive Response				
Total	0.463	0.066	7.027	0.000
Direct effect	0.263	0.076	3.469	0.001
Depression --> Anxiety --> Response defensive				
Specific indirect effect	0.200	0.054	3.704	0.000
Depression --> Stress				
Total	0.409	0.068	6.034	0.000
Direct effect	0.258	0.082	3.159	0.002
Depression --> Anxiety --> Stress				
Specific indirect effect	0.151	0.054	2.804	0.005
Extraversion --> Response defensive				
Total	−0.198	0.067	−2.928	0.003
Direct effect	−0.137	0.061	−2.238	0.025
Extraversion --> Anxiety --> Response defensive				
Specific indirect effect	−0.060	0.022	−2.238	0.006

## Data Availability

The datasets generated during and/or analyzed during the current study are available from the corresponding author on reasonable request.

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
