# Peer review of "The Route of Stress in Parents of Young Children with and without Autism: A Path-Analysis Study"

_ijerph, 2021, doi:10.3390/ijerph182010887_

Round 1

Reviewer 1 Report

This is an interesting study. However, its presentation lack some important information and needs some revision.

Abstract - "... a large sample of 165 parents (71 fathers and 94 mothers).

The sample is not so large due to the number of items used in the questionnaires. You say in the text that “Power calculation analysis revealed that our sample size was sufficient to reach a statistical power of 95 % (5% risk of type II errors) at a significance level of α < 0.05 with an estimated effect size of 0.9.” How was this power calculation performed? This is important for the robustness of the model.

Page 2 – Line 91 – “Therefore, a final sample of 165 parents (71 fathers and 94 mothers) was analyzed in 91 the study.”

It is important to know if you have used married people, because they are related in their opinions due to the fact that they have the same children.

Page 3 – Line 99 – “The PSI-SF is a 36-item, self-report measure designed to assess 99 parental distress in parents of children aged between birth and 12 years [46].”

Why do you not present children’s age mean and range? The sample of parents should be chosen also based in that characteristic.

Page 4 – Lines 154-155 – “The main model testing our hypotheses was: might psychological and demographic factors impact the perception of stress level and general health status in ASD and TD parents?”

This seems to be a research question rather than a description of the model.

Variable measures.

It would be better to say the type of answers used within each scale items and other variables used on this study.

Table 1.

Education father and Education mother were measured with an ordinal scale. The legend does not mention that the values are medians, instead of means.

Did you make an independent-sample t-test between those levels? I think there is a mistake in the symbol used is these variables.

Path analysis

It would be better to present first the model with a figure, and then explain what was done to study it. Presenting in the end the final model.

Figure 1 is far away from the text that mention it.

Page 5 – Lines 180-181 – “However, only gender, β = - .273, influenced anxiety. In other words, females predicted the levels of anxiety”.

It seems to me that this is not well explained. Probably, what you want to say is that females present higher levels of anxiety. What predicts anxiety is the gender and not only females.

Results

You present several variables included in the model that are not mentioned or clearly explained in their relationships in the literature review. Defensive response and openness are examples. Besides that, there a lack of definition of many of the concepts used in the model. Please, enhance your introduction that includes the literature review.

Page 8 – Lines 225-226 – “In this study, we provide for the first time a conceptual model to shed new light on the complex interaction between stress, psychological and personality traits, considering gender, in parents of children with and without autism.”

Is this new conceptual model based on the literature? If so, then this must be clear in the introduction and literature review. Do you propose new relationships between these variables that were not studied before? Then, you should be clear about this new contribution to science.

Good work.

Author Response

Reviewer n° 1

This is an interesting study. However, its presentation lacks some important information and needs some revision.

  • Abstract - "... a large sample of 165 parents (71 fathers and 94 mothers). The sample is not so large due to the number of items used in the questionnaires. You say in the text that “Power calculation analysis revealed that our sample size was sufficient to reach a statistical power of 95 % (5% risk of type II errors) at a significance level of α < 0.05 with an estimated effect size of 0.9.” How was this power calculation performed? This is important for the robustness of the model.

REPLY:  We thank the reviewer for this observation. Power analysis was made using the GPower tool. This additional information has been included in the methods section.

  • Page 2 – Line 91 – “Therefore, a final sample of 165 parents (71 fathers and 94 mothers) was analyzed in 91 the study.” It is important to know if you have used married people, because they are related in their opinions due to the fact that they have the same children.

REPLY:  Thank you for this comment. We have added this information in the Paragraph 2.1 as following: ‘In the ASD group n=56 couples were married, n=3 were divorced and n=1 was a single mother. In the TD group, n= 51 couples were married, n=1 couple was divorced and n=1 was a single mother’.

We also double-checked and corrected the information related to the original sample size, since it was wrongly reported from a previous draft.

3) Page 3 – Line 99 – “The PSI-SF is a 36-item, self-report measure designed to assess 99 parental distress in parents of children aged between birth and 12 years [46].” Why do you not present children’s age mean and range? The sample of parents should be chosen also based on that characteristic.

REPLY:  We thank the Reviewer for raising this question and we agree that children’s age is relevant. All the chidren were preschoolers, aged from 3 to 6 years old. We added this information as well as children’s sex ratio in the text, at Paragraph 2.1, as following:

‘A large  sample of 291 parents of young children with and without autism aged between 3 and 6 years of age, both males and females (25% females in the ASD group and 51% females in the TD group)’.

  • Page 4 – Lines 154-155 – “The main model testing our hypotheses was: might psychological and demographic factors impact the perception of stress level and general health status in ASD and TD parents?” This seems to be a research question rather than a description of the model.

REPLY:  We agree with the Reviewer and we rephrased the sentence as following: ‘The main model aimed to test whether demographic factors and psychological measures impacted the perception and levels of stress in parents of children with and without ASD.’

  • Variable measures. It would be better to say the type of answers used within each scale item and other variables used in this study.

REPLY:  Thank you for this suggestion. Now we have reported, for each instrument, how each item was measured.

  • Table 1. Education father and Education mother were measured with an ordinal scale. The legend does not mention that the values are medians, instead of means. Did you make an independent-sample t-test between those levels? I think there is a mistake in the symbol used is these variables.

REPLY:  Father  and mother education has been changed in total years of education being converted into a continuous variable (means and SD). An independent-sample t-test has been applied.

  • Path analysis: It would be better to present first the model with a figure, and then explain what was done to study it. Presenting in the end the final model. Figure 1 is far away from the text that mentions it.

REPLY:  Thank you for this suggestion. Now we have moved Figure 1 in the main text. The section “Path analysis” was revised.

  • Page 5 – Lines 180-181 – “However, only gender, β = - .273, influenced anxiety. In other words, females predicted the levels of anxiety”. It seems to me that this is not well explained. Probably, what you want to say is that females present higher levels of anxiety. What predicts anxiety is the gender and not only females.

REPLY:  Thank you for this suggestion. We have now improved the description of the results section as follows: “ However, only females gender, β = - .273, showed higher levels of influenced anxiety. In other words, gender females predicted the levels of anxiety”.

  • Results: You present several variables included in the model that are not mentioned or clearly explained in their relationships in the literature review. Defensive response and openness are examples. Besides that, there is a lack of definition of many of the concepts used in the model. Please, enhance your introduction that includes the literature review.

            REPLY:  We thank the reviewer for this suggestion. We have added a clear definition of the concepts used in the model in the Paragraph 2.2 Procedures, as well as added more literature review in the introduction (in relation to parental stress, anxiety, depression and socio-demographic features).

  • Page 8 – Lines 225-226 – “In this study, we provide for the first time a conceptual model to shed new light on the complex interaction between stress, psychological and personality traits, considering gender, in parents of children with and without autism.” Is this new conceptual model based on the literature? If so, then this must be clear in the introduction and literature review. Do you propose new relationships between these variables that were not studied before? Then, you should be clear about this new contribution to science.

REPLY:  We would like to thank this reviewer for this important suggestion. We have added literature related to the conceptual model in the introduction and have reformulated this sentence accordingly.

Reviewer 2 Report

Thank you for inviting me to review the paper entitled "The route of stress in parents of young children with and without autism: A path-analysis study". This is an interesting study examining the complex interaction between stress, psychological predisposition, and personality traits in parents of children with and without ASD by means of structural equation modeling. The authors found that in parents of typically developing children (TD), depression and age are the main direct predictors of stress through the mediating effect of anxiety. Conversely, the personality trait ‘openness’ directly predicts the defensive response and stress levels in parents of children with ASD, without the mediating effect of anxiety. As underlined by the authors, the findings of this study are relevant as they may suggest the promotion of new behavioral strategies to prevent parenting stress.

I compliment the authors for the original idea and the application of novel statistical methods to understand the complex interplay between psychological/psychopathological variables and parenting stress in ASD.

I have a few comments that may help further improve the quality of the manuscript.

Major comment/point to clarify:

  • Big Five: It is important to clarify whether emotional stability or neuroticism have been evaluated. They are the opposite and, as far as I know, neuroticism is commonly used as the "reference" trait in this questionnaire. Please, carefully check if the scores have been reversed or if there have been any errors in the scoring because it might change the results. Otherwise, clarify appropriately. 

Minor comments:

  • Abstract: line 21, I'd suggest replacing "autism" with "autism spectrum disorder (ASD)"
  • Line 83-84: "autism spectrum disorder" already defined; please, replace with "ASD"
  • Lines 98-118: I think it might be clearer to separate the scales' descriptions in separated subparagraphs, using a list (i.e. start a new line for a new scale)
  •  Big Five Questionnaire: there are different versions. Did you use the 132 items versions? Self-report? Likert scale for answers? Please, describe more in detail. See also "Major comments".
  • Lines 154-156: I think that the expression "general health status" is inappropriate in this case, as one would expect for instance the administration of general well-being (e.g., GHQ-12) or the evaluation of physical conditions. I'd suggest changing this part with the specific psychopathological variables investigated (i.e. depression and anxiety).
  • Table 1: I don't think that reporting p-values and Cohen's d adds any meaningful information to this table, especially because it refers to non-significant differences. Therefore, I suggest removing the last two columns from Table 1 (you can just specify in the text that there are no significant differences). On the contrary, it would be important to report Cohen's d in Table 2, to estimate the actual magnitude of the difference in psychological/psychopathological variables between parents of children with and without ASD.
  • Table 1, Education mother and Education father: I think this data has been incorrectly reported. It should be represented as a categorical variable, while here it seems reported like a median and IQR which are for continuous variables. My suggestion is to report this information as a categorical variable or alternatively as continuous using the years of education instead of the highest educational level.
  • Table 1, Job mother and job father: please, remove the bullets from the table
  • Table 2: Mood status is in the same line as Depression. 
  • Table 2: please, replace .000 with <0.001
  • Table 2: please, specify that values are reported as means ± SD
  • Table 2: I suggest reporting the name of the tool that has been used to evaluate each of the domains). For instance, it is unclear what defensive response and perception refer to. (I guess they are the domain of the PSI but it is not clear, as other domains are reported in the descriptions in the MEthods section.
  • Discussion: I think in the discussion it would be important not only to repeat the findings of the study but also propose model of interventions that might help prevent mental health issues and support families of autistic individuals.
  • Limitations: As reported in lines 88-90, 43 parents were excluded from the sample (around 18% of the originally recruited sample) have been excluded from the analysis because of incomplete responses. This should be mentioned among limitations. 
  • Another limitation to mention is that only some psychological traits have been evaluated as potential predictors of psychopathology. Other potentially influencing variables (e.g. severity of ASD, age of children with/without ASD) have not been considered in the model while could impact the onset of depression and anxiety.
  • Finally, I suggest carefully check English language. 

Author Response

Thank you for inviting me to review the paper entitled "The route of stress in parents of young children with and without autism: A path-analysis study". This is an interesting study examining the complex interaction between stress, psychological predisposition, and personality traits in parents of children with and without ASD by means of structural equation modeling. The authors found that in parents of typically developing children (TD), depression and age are the main direct predictors of stress through the mediating effect of anxiety. Conversely, the personality trait ‘openness’ directly predicts the defensive response and stress levels in parents of children with ASD, without the mediating effect of anxiety. As underlined by the authors, the findings of this study are relevant as they may suggest the promotion of new behavioral strategies to prevent parenting stress.

I compliment the authors for the original idea and the application of novel statistical methods to understand the complex interplay between psychological/psychopathological variables and parenting stress in ASD.

I have a few comments that may help further improve the quality of the manuscript.

Major comment/point to clarify:

  • Big Five: It is important to clarify whether emotional stability or neuroticism have been evaluated. They are the opposite and, as far as I know, neuroticism is commonly used as the "reference" trait in this questionnaire. Please, carefully check if the scores have been reversed or if there have been any errors in the scoring because it might change the results. Otherwise, clarify appropriately. 

REPLY:  We are aware of the possible mistake and we are grateful to the Reviewer for pinpointing the possible confounder. We confirm that we scored Emotional Stability, which is the label we used consistently in the text.

Minor comments:

  • Abstract: line 21, I'd suggest replacing "autism" with "autism spectrum disorder (ASD)"

REPLY:  Done.

  • Line 83-84: "autism spectrum disorder" already defined; please, replace with "ASD"

REPLY:  Done.

  • Lines 98-118: I think it might be clearer to separate the scales' descriptions in separated subparagraphs, using a list (i.e. start a new line for a new scale)

REPLY:  Done.

  • Big Five Questionnaire: there are different versions. Did you use the 132 items versions? Self-report? Likert scale for answers? Please, describe more in detail. See also "Major comments".

REPLY:  We thank the reviewers for this Comment. We used the Big Five version with 132 self-assessment elements. Items were rated on a five-point Likert scale ranging from 1 = (very false for me) to 5 = (almost always true). We added these information in Paragraph 2.2 Procedures.

  • Lines 154-156: I think that the expression "general health status" is inappropriate in this case, as one would expect for instance the administration of general well-being (e.g., GHQ-12) or the evaluation of physical conditions. I'd suggest changing this part with the specific psychopathological variables investigated (i.e. depression and anxiety).

REPLY:  Thank you for this suggestion. We have changed accordingly.

  • Table 1: I don't think that reporting p-values and Cohen's d adds any meaningful information to this table, especially because it refers to non-significant differences. Therefore, I suggest removing the last two columns from Table 1 (you can just specify in the text that there are no significant differences). On the contrary, it would be important to report Cohen's d in Table 2, to estimate the actual magnitude of the difference in psychological/psychopathological variables between parents of children with and without ASD.

REPLY:  Following reviewer’s suggestion Cohen's d values have been removed from table 1 and inserted in Table 2.

  • Table 1, Education mother and Education fatheReply: I think this data has been incorrectly reported. It should be represented as a categorical variable, while here it seems reported like a median and IQR which are for continuous variables. My suggestion is to report this information as a categorical variable or alternatively as continuous using the years of education instead of the highest educational level.

REPLY:  We thank the Reviewer for the suggestion and have reported education as a continuous variable, using the years of education and rerunning the correct stats.

  • Table 1, Job mother and job fatheReply: please, remove the bullets from the table

REPLY:  Done.

  • Table 2: Mood status is in the same line as Depression. 

REPLY:  Done.

  • Table 2: please, replace .000 with <0.001

REPLY:  Done.

  • Table 2: please, specify that values are reported as means ± SD

REPLY:  Done.

  • Table 2: I suggest reporting the name of the tool that has been used to evaluate each of the domains). For instance, it is unclear what defensive response and perception refer to. (I guess they are the domain of the PSI but it is not clear, as other domains are reported in the descriptions in the MEthods section.

REPLY:  We are grateful to the Reviewer for addressing this issue. We have checked and corrected in the text and tables the inconsistent labels that should now be all clear.

  • Discussion: I think in the discussion it would be important not only to repeat the findings of the study but also propose model of interventions that might help prevent mental health issues and support families of autistic individuals.

REPLY:  We thank the Reviewer for this comment. We have added this aspect related to intervention and support to family in the latter paragraph of the Discussion.

  • Limitations: As reported in lines 88-90, 43 parents were excluded from the sample (around 18% of the originally recruited sample) have been excluded from the analysis because of incomplete responses. This should be mentioned among limitations. 

REPLY:  Thank you for this advice. We have now reported this information in the limitations’ paragraph.

  • Another limitation to mention is that only some psychological traits have been evaluated as potential predictors of psychopathology. Other potentially influencing variables (e.g. severity of ASD, age of children with/without ASD) have not been considered in the model while could impact the onset of depression and anxiety.

REPLY:  Thank you for this remark. We have considered all variables in the study, pooling them together in our path-SEM analysis. Interestingly what could seem important influencing variables (e.g. severity of ASD as the reviewer truthfully noted) did not show in our study significant correlations with perception of stress and defensive response. 

  • Finally, I suggest carefully check English language. 

REPLY:  An additional English editing was performed

Round 2

Reviewer 2 Report

The authors have adequately addressed my comments and the manuscript is substantially improved. Good job!